# An Investigation on the Efficacy of Orotic Acid as a Bio-Nucleating Agent for Poly-Lactic Acid under Quiescent Condition and Injection Molding

**DOI:** 10.3390/mi13122186

**Published:** 2022-12-10

**Authors:** Peng Gao, Davide Masato, Animesh Kundu, John P. Coulter

**Affiliations:** 1Manufacturing Science Laboratory, Department of Mechanical Engineering and Mechanics, Lehigh University, Bethlehem, PA 18015, USA; 2Department of Plastics Engineering, University of Massachusetts Lowell, Lowell, MA 01854, USA

**Keywords:** additive, nucleation, crystallization, modeling, DSC

## Abstract

Polylactic acid (PLA) is a bio-based biodegradable polymer and is considered to be an environmentally friendly alternative to petroleum-based polymers for various applications. Neat PLA requires an extended period at elevated temperatures to attain its maximum crystallinity, which can be mitigated by the addition of nucleating agents. Orotic acid is a natural heterocyclic nucleating agent in PLA. The effect of orotic acid on the crystallization behavior of a commercial, high-purity PLA was studied in detail. A differential scanning calorimetry (DSC) technique was utilized for this purpose. A new protocol for the quantitative characterization of crystallization kinetics from DSC data was developed. It was found that the total crystallinity increased from 26% to 63% at 80 °C with 1% content of orotic acid. Meanwhile, the crystallization rate of PLA-OA blends increased by ~10 times as compared to neat PLA. The addition of orotic acid also reduced the incubation time by >17% under quiescent conditions. Injection molding experiments showed that highly crystallized (>50%) PLA samples could be fabricated with a 1% addition of orotic acid. The required mold temperature was reduced from the 120 °C recommended by the supplier to 80 °C.

## 1. Introduction

### 1.1. PLA and Additives for PLA

PLA is a bio-based and biodegradable thermoplastic polymer derived from renewable resources, in contrast to common commercial-grade thermoplastics, such as those from the polyethylene family, and isotactic polypropylene, which are derived from nonrenewable petroleum reserves [1].

PLA refers to a family of materials each having slightly different characteristics. Lactic acid has two stereoisomers namely the L-lactic and the D-lactic acid resulting in three forms of PLA that are available commercially: pure L-lactide, pure D-lactide, and a mix of L and D-lactide. Relatively pure L-feed and D-feed PLA are referred to as PLLA and PDLA respectively [2]. Typical commercial-grade PLA with high crystallinity contains a majority of L-feed containing ~1–2% D content, whereas the amorphous grades may contain up to 20% D feed. The crystallinity and other physical properties of these materials can be varied during the processing of these materials by utilizing different processing conditions. Additives, such as nucleating agents [3], accelerants, impact modifiers, and mold flow agents, are added to the commercial grades to affect the processibility, crystallinity development, and other properties, particularly mechanical properties, in these materials. Harris et al. showed that PLA with 20% crystallinity could achieve a 20% higher flexural modulus compared to amorphous PLA of the same grade. As the crystallinity increased to 40%, the flexural modulus increased by 25% [4]. However, the crystallization rate of PLA is slow, thus obtaining sufficiently high crystallinity within reasonable manufacturing times is difficult.

The crystallinity of PLA products can be intrinsically enhanced by employing such methods as isothermal annealing, polymer blending, and strain-induced crystallization. Isothermal annealing at temperatures in the range of 85–115 °C for an extended period has been reported to initiate and develop the crystalline domains [5]. The addition of nucleating agents is effective in enhancing the crystallinity in PLA [6,7,8,9,10,11,12,13,14,15]. These additives can be organic, such as poly (ethylene/butylene succinate), also known commercially as Bionolle, the potassium salt of 3,5-bis(methoxycarbonyl)-benzenesulfonate (LAK-301), or inorganic, such as talc. It has been reported that the half-crystallization time(t_1/2_) at 110 °C can be reduced from 25 min to 0.4 min by adding 1 wt% of talc to a 99/1 PLLA/PDLA blend [16]. Stereocomplex formed by equimolar mixtures of PLLA and PDLA was reported to reduce the half-crystallization time (t_1/2_) at 140 °C from 17–20 min to less than 1 min [17]. A small amount of 3 wt% of the stereocomplex was sufficient to reduce the half crystallization drastically.

In addition to increasing the crystallization rate, the nucleating agents can reduce the energy barrier for nucleation and effectively control the crystal morphology of certain semi-crystalline polymers including PLA [18]. It has been demonstrated that N,N′,N′′-tricyclohexyl-1,3,5-benzenetricarboxylamide (TMC-328), a small molecular organic nucleating agent, can dissolve in PLA melt during fabrication and self-organize into fine fibrils during cooling stage, inducing the epitaxial growth of PLA lamellae and formation of a shish-kebab like structure [16].

### 1.2. Orotic Acid as a Nucleating Agent for PLA

To preserve the benefit and advantages of PLA, the additives for commercial PLA must be non-toxic and biodegradable. Even though talc and C-60 have been proven to be efficient nucleating agents of PLA, these inorganic additives are replaced by organic additives, such as orotic acid and LAK series [16].

There have been limited studies on the efficacy of orotic acid (OA) as a nucleating agent. The reported results are often contradictory. Feng et al. observed that orotic acid is minimally effective in promoting crystallization in an optically pure PLLA [16]. Only 14% crystallinity was reported in a 0.5 % (*w/w*) PLA-OA blend at 120 °C. In contrast, Salač et al. reported OA to be an effective nucleating agent for PLA2002D from NatureWorks [11]. In a non-isothermal cooling experiment, they observed 31% crystallinity in a 0.5% (*w/w*) PLA-OA blend. In yet another study, Qiu et al. reported OA increased the crystallization kinetics of PLLA dramatically by about an order of magnitude in the temperature range 120–140 °C. The discrepancy in the literature can be largely attributed to the different grades of PLA utilized in those research works, as well as the employed analysis procedures.

### 1.3. Injection Molding of PLA Products

Injection molding process is the most widely used technique in PLA product fabrication. During the injection molding process, a polymer melt is forced into a small channel such as the nozzle of runner system during injection molding, it is subjected to high shear stresses. Since the semi-crystalline polymer melt is solidified under a stressed state, a combination of crystallization by stretching and melt solidification is expected to affect the crystallinity development. PLA, particularly, requires both the assistance of nucleating agent and proper injection molding processing parameters to enhance the crystallinity. The datasheet suggests that to mold PLA 3100HP requires 1% LAK-301, which is a powder-based nucleating agent, at 120 °C mold temperature for 3 min to ensure high crystallinity.

The efficiency of orotic acid as a nucleating agent of PLA under quiescent conditions and using injection molding technology was investigated in this research by utilizing Differential Scanning Calorimetry (DSC) technique. A relatively purer grade of commercial PLA was selected for the research. The experimental data for quiescent conditions were critically analyzed using Avrami crystallization kinetics. A novel protocol for quantitative characterization was developed. The degree of crystallinity, crystallization rate, and incubation time were compared between neat PLA and PLA-OA blends at various isotherm temperatures. The degree of crystallinity was obtained and compared on samples fabricated with neat PLA and PLA-OA blends under different processing conditions.

## 2. Sample Preparation and Characterization Technique

### 2.1. Material Selection

An Ingeo 2500HP PLA from NatureWorks was utilized in this study. This particular grade of PLA contains PLLA blended with 2% of PDLA and can be molded into semi-crystalline samples based on datasheets provided by NatureWorks. The material was in pellet form originally. The pellets were dried at 40 °C for 8 h to reduce the amount of absorbed moisture as instructed.

To enhance crystallinity and understand the effects of DISC-IM on PLA crystallization, 97% anhydrous orotic acid (OA) from Alfa Aesar was utilized as a nucleating agent for neat 2500HP PLA. The dried neat PLA was mixed with 1 wt.% of nucleating agent and processed utilizing a Brabender single screw extruder at 190 °C nozzle temperature. The processed filaments had a diameter of 1 mm. It was then chopped to 2 mm length for further characterization and fabrication. These samples are referred to as PLA-OA blends in the following sections.

### 2.2. Injection Molding Processing Parameters

The effect of orotic acid during PLA injection molding process was investigated with dog-bone samples fabricated with Nissei PS40E5A injection molding machine with a NC9000G controller. The samples fabricated were ASTM D638 Type I dog-bone samples, 165 mm in length, 13 mm in gage width, 50 mm in gage length, and 3 mm in thickness. The mold temperature was maintained by a temperature control unit. Key processing conditions are presented in Table 1. Five samples were collected using each processing condition.

Figure 1 presents samples fabricated with neat PLA and PLA-OA blends, respectively. Figure 1a shows the neat PLA sample fabricated at a mold temperate of 40 °C, while Figure 1b presents the PLA-OA sample at a mold temperature of 90 °C. For neat PLA, the sample is soft and warped even with the maximum allowed cooling time at mold temperatures above 60 °C. When fabricated using 40 °C mold temperature, the sample can be solidified within 30 s of cooling time. However, the sample was transparent when mold opened, indicating low crystallinity. It was observed that rigid PLA-OA dog-bone samples could be achieved at 60 °C and 80 °C in semicrystalline form with cooling time of 180 s or longer. Reducing cycle time led to soft samples that warped upon mold opening. To eliminate the effect of different cooling times, the samples were held in the mold for 180 s regardless of the optimal cooling time needed to ensure defect-free fabrication. The soft and warped samples were discarded. Only the samples fabricated with neat PLA at 40 °C mold and with PLA-OA blends at 80 °C and 90 °C mold were collected and prepared for further analysis.

### 2.3. Characterization Techniques

#### 2.3.1. Isothermal DSC Characterization Technique

The crystallization behavior of neat PLA and PLA-OA blends under quiescent conditions was studied. Differential scanning calorimetry was utilized to gain a fundamental understanding of the phase transitions and the melting behavior of PLA-OA blends. Of particular interest was the effect of orotic acid on the crystallization of PLA. A Q2000 DSC system from TA Instruments was utilized for this purpose. Samples weighing approximately 8–10 mg of dried neat 2500HP PLA pellet and chopped PLA-OA blend filaments were utilized for each of the DSC experiments involving isothermal heating at various temperatures for extended periods. The isothermal test protocol utilized was identical for all the experiments and consisted of the following steps.

Cycle 1: Heating a sample from 25 °C to 240 °C at 20 °C/min followed by an isotherm at 240 °C for 3 min. This cycle was intended to melt and remove all thermal history of the pellets. The 3-min isotherm ensured the complete melting of the sample.

Cycle 2: Cooling the material rapidly using the maximum cooling rate at 60 °C/min to various isotherm temperatures (80 °C, 90 °C, 100 °C, and 120 °C) to minimize any non-isothermal behavior during this cooling cycle.

Cycle 3: The material was isothermally held at the abovementioned isotherm temperatures for 30–120 min. The crystallization process was highly influenced by the isotherm temperature. To capture the full crystallization process in the isotherm cycle, an optimal isotherm time was provided for each temperature.

Cycle 4: On completion of the isotherm, the material was heated at 10 °C/min to 240 °C. The degree of crystallinity achieved from the isotherm was quantified from the melting peak observed during this heating cycle. The degree of crystallinity (*X_C_*) was calculated using Equation (1).
(1)$$X_{C}=\frac{\Delta H_{m}-\Delta H_{c}}{\Delta H_{M}}\times 100$$
where $\Delta H_{m}$ is the melting enthalpy (J/g), $\Delta H_{c}\;$ is the cold crystallization enthalpy (J/g), and $\Delta H_{M}$ is the melting enthalpy of a PLA crystal of infinite size, which was assumed to be 93 J/g in this research.

During cycle 2, a short period was required to stabilize at the isotherm temperature because of the thermal inertia of the system. As a result, the heat flow during the initial stages of the isotherm period was often representative of the system stabilization and the heat flow related to crystallization was completely masked, particularly for rapid crystallization events with a short incubation period. Such was the case for the PLA-OA blends but not for the neat PLA. No cold crystallization was observed during cycle 4 in these experiments.

#### 2.3.2. Dynamic DSC Characterization Technique

The melting behavior of injection molded samples investigated utilizing a Q2000 DSC system from TA Instruments. A cross sectional slice at the mid-section of the sample shown in Figure 2, weighing approximately 8–10 mg, was extracted and tested in the DSC system. The sample tested in DSC included both surface layers, intermediate layers, and the core of the PLA dog-bone specimen. Only the first heating scan performed at 10 °C/minute from 25 °C to 200 °C was collected to investigate effects of different molding parameters on the samples. The degree of crystallinity (*X_C_*) was calculated using Equation (1) above.

## 3. Data Treatment Technique

The heat flow data of PLA-OA blends during the isotherm cycle are presented in Figure 3. An overlap between the system stabilization and heat of crystallization was observed during the initial stages of the isothermal period. The degree of overlap was distinct for different isotherm temperatures. At lower temperatures, the crystallization rates were slow and the incubation time was longer, resulting in an insignificant overlap, as could be seen for the heat flow during the isotherm at 90 °C. A greater degree of overlap was observed as the isotherm temperatures approached the crystallization temperature (100–120 °C). At the higher temperatures, the incubation time was longer again because of the dominance of thermal vibrations that prohibited the formation of stable nuclei of critical size within a short time.

The incubation period and the initial part of the crystallization peak cannot be distinguished from the endothermic behavior due to this overlap while the system was stabilizing from the melt temperature (240 °C) to the isotherm temperature. A novel analysis method was conceived and successfully implemented to deconvolute the masked section of the crystallization peak and quantifiability of the isothermal crystallization of the PLA-OA blend. This is described in detail later in this section.

The kinetics of the crystallization process as observed from the DSC experiments was quantitatively studied utilizing the Avrami equation, as shown in Equation (2):(2)$$v_{c}=1-\exp\left[-K{\left(t\right)}^{n}\right]$$
where $v_{c}$ is the volumetric fraction of the converted phase at the time *t*; *K* is the crystallization rate constant, and *n* is the Avrami index.

However, the Avrami equation provides information about the total fraction crystallized as a function of time. The rate of heat generation (or absorption) as a function of time is measured in the DSC isotherm experiments. The DSC isotherms could be numerically integrated to calculate the total heat generation over a period of time. The crystallized fraction can then be determined experimentally if the enthalpy of crystallization for that particular phase is known. Alternately, the Avrami equation could be differentiated with respect to time to fit the theoretical model with the experimental data. In this research, a time derivative of the Avrami equation was utilized to quantify the experimental DSC data. Additional assumptions and modifications were needed to quantitatively assess the experimental data obtained from DSC utilizing the Avrami equation, as described in the following section.

I. Firstly, the Avrami equation is based on volumetric fraction whereas DSC data is typically normalized with mass. The relative volumetric fraction (*v*_c_) can be related to the mass fraction (*w*_c_) by Equation (3).
(3)$$v_{c}=\frac{w_{c}}{w_{c}+\frac{\rho_{c}}{\rho_{a}}\left(1-w_{c}\right)}$$
where *ρ_c_* and *ρ_a_* are the densities for fully crystallized and fully amorphous polymer. In this research, the densities were assumed to be similar. Hence, the weight fraction can be assumed to be similar to the volume fraction. We estimated that this can lead to an error in the absolute values by <10%, but the relative trends measured were quite accurate.

II. The total heat flow per unit mass during crystallization, Q, measured at any given instant can be calculated as
(4)$$Q=w_{c}\cdot \Delta H_{c}=v_{c}\cdot \Delta H_{c}$$
where $\Delta H_{c}$ is the enthalpy of crystallization per unit mass.

The Avrami equation can be modified utilizing Equation (4) as
(5)$$Q=\Delta H_{c}\cdot \left(1-\exp\left[-K{\left(t\right)}^{n}\right]\right)$$

III. The Avrami equation describes the crystallization process after it has been initiated and does not account for the incubation time. The incubation time is highly dependent on the temperature. Certain flexible polymers, such as polyethylene, whose nucleating process is very fast will start to crystallize immediately upon thermal equilibrium at the crystallization temperature. In contrast, semi-flexible polymers such as polyesters would need several minutes for the crystallization process to start. To account for the incubation period another time parameter, a time term, *t*_0_ was introduced in the Avrami equation as follows.
(6)$$Q=\Delta H_{c}\cdot \left(1-\exp\left[-K{\left(t-t_{0}\right)}^{n}\right]\right)$$

The experimental time, *t*, was set to zero at the beginning of the isothermal heat treatment period. The value of *t_0_* can be positive or negative. A positive value indicates that the crystallization process started later after the system was equilibrated at the isotherm temperature, while a negative value indicates that the crystallization process started at the previous stage when the material was cooling from the melt to isotherm temperature.

The time derivative of the modified Avrami equation (Equation (6)) can be presented as
(7)$$\frac{dQ}{dt}=\Delta H_{c}\cdot K\cdot \exp\left[-K{\left(t-t_{0}\right)}^{n}\right]\cdot n\cdot {\left(t-t_{0}\right)}^{n-1}$$

The heat flow rate, $\frac{dQ}{dt}$ was experimentally obtained from the DSC isotherm data as a function of time and was utilized to determine the constants, $\Delta H_{c},\;\;t_{0},$
*k*, and *n*.

The exact process for utilizing the modified Avrami equation in Equation (7) is elucidated utilizing the experimental data for PLA-OA at 90 °C isothermal temperature. The time was set to zero to mark the start of the isotherm as shown in Figure 4. As previously discussed, the system could not be instantaneously cooled down and stabilized at the isothermal temperature during the experiments because of the thermal inertia of the system. As a result, the heat flow during rapid cooling and stabilization at the isothermal temperature overlaps with the crystallization exotherm and cannot be deconvoluted numerically. The temperature eventually stabilized at ±0.05 °C of the isotherm temperature during the isothermal hold. The data for the period at the beginning of the isothermal hold were considered unreliable and masked for further analysis.

There has been a long debate concerning the form of experimental baseline before, during, and after isotherm crystallization captured by DSC. Several procedures have been proposed to establish the baseline correctly. The most commonly used method involves utilizing a horizontal line that is representative of the heat capacity of the material [19]. In this research, to minimize the effect of system noises, a linear baseline of a constant value was derived from the isotherm cycle. The inherent assumption was that the crystallization process was complete towards the end of the isotherm cycle. In some samples, fluctuations of ±0.0002 mW were observed near the end of the isotherm. Based on the accuracy and resolution of the measurements, such values were determined to be noise and only three digits after decimal points were considered for further analysis. The minimum heat flow value was used as the baseline and subtracted from the DSC data for further analysis in this research.

It is critical to choose a reasonable conversion range from the baseline-corrected data for fitting the modified Avrami equation. In most cases, the initial data points were not reliable and need to be neglected due to the stabilization of the system. The secondary crystallization process or other exothermic processes can produce a broad shoulder at longer times in the DSC isotherm data. Lorenzo et al. investigated the effect of the selected range and concluded that the entire range should not be utilized for deriving the Avrami constants, *t,* and *n*. The experimental data for conversion up to 20% was determined to be optimal [19]. Accordingly, a suitable section of the data was selected for curve fitting utilizing the modified Avrami equation. Representative fitted data are presented in Figure 4. The coefficients determined from best fit were utilized to create the initial section of the DSC isotherm data, where the experimental data were affected by stabilization of the system, and hence determined to be unreliable for investigating crystallization kinetics. An example of the hybrid experimental-computed heat flow curve is presented in Figure 5, where the experimental data are marked in black and the computed data in red.

The total enthalpy for the crystallization process was determined by integrating the area of the hybrid curve, the cumulative curve is also shown in Figure 5. The degree of crystallinity was quantitively determined from the melting peak that was observed on subsequent melting of the crystallized fraction during the DSC experiments as presented earlier. The melting enthalpy for this grade of PLA was 93 J/g. The isothermal data were normalized with the crystallized fraction data from the melting peak and presented in traditional Avrami plots.

## 4. Result and Discussions

### 4.1. Overall Crystallinity Development under Quiescent Conditions

The crystallinity for PLA-OA blends and neat PLA as calculated from the melting peak of the crystallized fraction in cycle 4 is presented in Table 2. The uncertainty in the measurements was determined to be 7.5%. The melting peaks were often broad and overlapping melting peaks of the α and α’ phases PLA were observed in almost all samples. As a result, the choice of the starting point of the melting peak and consequently the resultant background curve for the melting peak were not straightforward, and this was the primary reason for the uncertainty in measurements.

Regardless of the uncertainty, it could be observed that the total crystallinity increased with increasing temperature for both neat PLA and PLA-OA blends in general. For neat PLA, the crystallinity was 26% at 80 °C and increased to a maximum of 55% at 120 °C.

The most dramatic differences in crystallinity development between PLA-OA blends and neat PLA were observed at the lowest and highest temperatures. At 80 °C, when the enthalpy provided is low, the total crystallinity in the PLA-OA blend was 63% as compared to 26% for neat PLA. The results indicate that orotic acid is an effective nucleating agent. It was surmised that orotic acid not only provided additional nucleation sites, but reduced the activation barrier as well.

### 4.2. Crystallization Behavior under Quiescent Conditions

The heat-flow data obtained from the isotherm cycle for samples are presented in Figure 6. The total enthalpy associated with the crystallization during the isotherm was numerically calculated from the heat flow data presented in Figure 7 and normalized with the degree of crystallinity obtained from the melting peak to determine the crystallinity development as a function of time. The results for different isotherm temperatures are presented in Figure 6. The current data analysis procedure incorporated a time term (*t*_0_) in the Avrami equation. This provided a convenient means to mathematically determine the starting of the crystallization process and thus compare the incubation time. The origin of the time scale (*t* = 0) in Figure 6 and Figure 7 was shifted accordingly.

The corresponding incubation times and initial crystallization slope values are presented in Table 3. For both the materials, neat PLA and PLA-OA blend, the incubation time decreased with the increasing temperature, reaching a minimum value before increasing with temperature. At lower temperatures, the enthalpy provided was low for the formation of critical nuclei, while at higher temperatures the Brownian motion disrupted the nucleation process, resulting in a longer incubation time at these temperatures. For neat PLA, the minimum incubation time, 2.63 min, was at 100 °C, while that for PLA-OA blend (2.34 min) was at 120 °C. The incubation time for PLA-OA blends was shorter than that for neat PLA at any given temperature. This could be attributed to the additional nucleation sites provided by the nucleating agent.

Crystallization is a first-order phase transition with an enthalpy associated with the transition and the rate of heat evolution is a measure of the rate of crystallization. The exothermic peaks for neat PLA, presented in Figure 6a, were broad and required a much longer time (120 min) before the heat flow curve was linear. At 100 °C, a distinct exothermic peak was observed for this sample and the heat evolution rate was minimal after ~35 min, indicating that the rate of crystallization is slow after that time. In contrast, at 120 °C, the broad exothermic peak was followed by a long tail/shoulder. The growth of the crystalline regions probably contributed to the longer exothermic tail in the isotherm peaks.

For neat PLA, the total crystallinity that was developed during the isotherm period (120 min) was similar (53.5 ± 0.5%) for temperatures between 90 °C and 120 °C, while at 80 °C, the total crystallinity was only 26% after 120 min. However, the crystallization rate was distinctly different at the different isotherm temperatures. The crystallization rate was fastest at 100 °C but the total crystallization was maximum at 120 °C by a small margin. Even at the fasted rate at 100 °C, 25% crystallinity was achieved in approximately 800 s (13.3 min) under quiescent conditions. An additional 700 s (11.7 min) was required to reach the same degree of crystallinity at 120 °C. At 90 °C, the time required increased to ~2200 s (36.7 min) and at 80 °C, 25% crystallinity was almost the maximum that PLA could reach this temperature. It required 4500 s (75 min) to reach 25% crystallinity. All these results indicate that although the neat 2500HP PLA is capable of crystallizing to a high degree (>50%) in the temperature range of interest, the crystallization rates are slow, and the incubation time is long. This grade of PLA will potentially be an ineffective material for industrial manufacturing such as injection molding with high crystallinity by itself.

The crystallization rates were extremely fast during the initial stage of the crystallization process of PLA-OA blends at temperatures ≥ 90 °C in contrast (Figure 7b). Additionally, the initial crystallization rate was almost identical at 100 °C and 120 °C, as indicated by the slope of the initial sections of the respective isotherm curves. A total crystallized fraction of 25% was developed within the first 45 s after the formation of stable nuclei (incubation period) at these temperatures. The fast crystallization could be attributed to the formation of a large number of stable nuclei during the initial stages of the crystallization aided by orotic acid. However, it is not clear how a small amount of orotic acid (1%) could result in such a dramatic increase in crystallinity in a short period.

The crystallization kinetics of neat PLA and PLA-OA blend in the temperature range of 80–120 °C are compared in Figure 8. The initial crystallization rates are indicated in the figure. The crystallization kinetics plots are obtained by differentiating the total crystallization curves. The incubation time was ignored for direct comparison. As expected, the shape of the curves is identical to the heat flow curves experimentally derived sans the shift in the time scale because of the incubation time and the designation of the y-axis. The most significant observation was that 1 wt.% orotic acid reduced the time required to achieve similar or even higher crystallinity from 120 min to 30 min. It was also observed that the addition of orotic acid not only enhanced the crystallization kinetics by an order of magnitude, but also resulted in higher total crystallinity in the materials.

The total crystallinity increased for PLA-OA blends as compared to neat PLA samples at the temperature range from 80 °C to 120 °C. The trends observed in this research are largely in agreement with observations by other researchers. Y. Feng et al. observed that PLLA-L130 increased the crystallinity from nonexistent (0%) for neat PLLA to 13.9% for 0.5% (*w/w*) nucleating agent mixed PLLA-OA blend at the isotherm temperature of 120 °C for 30 min [16]. Other nucleating agents, such as LAK-301 (potassium salt of 3,5-bis(methoxycarbonyl)benzenesulfonate), TMC-306 (substituted-aryl phosphate salts (TMP-5), N’1,N’6-dibenzoyladipohydrazide), and talc, can further increase the total crystallinity to a maximum of 45.7% under identical isotherm conditions [16]. J. Salač et al. observed the same trend in a non-isotherm crystallization study. The crystallinity increased from 0% for neat PLA 2002D to 34% for 5.0% (*w/w*) PLA-OA blend [11]. Further increasing the concentration of nucleating agents did not impact the crystallization kinetics. Although the trends are similar, orotic acid was observed to be more effective than reported in the literature. The total crystallinity obtained in PLA on doping with orotic acid in this research was significantly larger. This could be attributed to the different grades of PLA utilized by different research groups. The different grades could contain different additives.

It was surmised that the optical purity of the PLA materials had the most significant effect on the crystallization rates as well as the total crystallization observed. A higher D-lactide concentration is known to hinder the crystallization behavior of PLA materials [20]. In most cases, D-lactide content was not reported. In this research, the initial crystallization rate increased by ~30 times when 1% orotic acid was added to the PLA as shown in Figure 8 and Table 3. The initial crystallization rates were observed to be similar in a study performed by Qiu et al. for a relatively pure PLLA material [10]. In their research, the total time required for PLLA to fully crystallize decreased from ~20 min for neat PLLA to less than 2 min for PLLA-0.3% OA blend at an isotherm temperature of 120 °C. In contrast, neat PLA 2500HP fully crystallized in ~70 min in this study, and PLA-1%OA fully crystallized in ~10 min at 120 °C. The grade utilized in that research was a neat PLLA, and the concentration of D-lactide is postulated to be less than 1%. The PLA 2500HP utilized in the current research contained approximately 2% of D-lactide. Only the relative crystallization data were reported by Qiu et al., so it was not possible to compare the total crystallization obtained in these similar blends [10].

The *k*, *n*, and $\Delta H_{c}$ values obtained from the Avrami heat flow curve fitting scheme are presented in Table 4. For neat PLA, the Avrami index showed very consistent values between 1.6 and 1.7, while the crystallization rate factor k varied between 1.08 × 10^−3^ and 3.72 × 10^−3^, and crystallization enthalpy $\Delta H_{c}$ varied from 0.35–0.90. The maximum crystallization rate value of *k* = 3.72 × 10^−3^ was observed at 100 °C and was significantly reduced to *k* = 1.08 × 10^−3^ and *k* = 1.09 × 10^−3^ at 80 °C and 90 °C isotherms. The $\Delta H_{c}$ values, which represent the total enthalpy per unit mass neat PLA crystallize, matched the trend of the degree of crystallization obtained from the second melting cycle. Even though the total enthalpy of 80 °C and 90 °C isotherm was significantly different, the initial crystallization rate during the first 20 min into the isotherm cycle was identical for the two runs. The difference on $\Delta H_{c}$ value was an outcome of different growth rates and the effects of secondary crystallization behaviors during the 120-min isotherm.

For PLA-OA blends, the fitted data at 80 °C were not reported due to weak isotherm crystallization behaviors. Additionally, the crystallization behavior was not completed during the 30-min test at 80 °C, so the total enthalpy $\Delta H_{c}$ value was not accurate. For the runs performed at 90–120 °C, the Avrami index varied from 1.2–1.7 while the crystallization rate factor, *k* varied between 0.18 to 0.57, and crystallization enthalpy, $\Delta H_{c}$ varied from 0.82–0.88. The maximum crystallization rate of 0.57 was observed at 100 °C and reduced to 0.18 at 90 °C. Compared to neat PLA, the crystallization rate of PLA-OA blends increased by two orders of magnitude at the same isotherm temperature, indicating that orotic acid is an effective nucleating agent for PLA isothermal crystallization. Additionally, the crystallization rate during the first 60 s after crystallization process initialized increased by ~2 times as compared to the average crystallization rate of the entire isotherm cycle. This indicated that the initial crystallization, dominated by the nucleation of potential nuclei, is significant. During the initial crystallization period, the density of nuclei increases dramatically, creating potential nucleation regions to develop and grow in the later stage. After the nucleation period, the more dominant behavior is the growth of crystal regions created during the initial period. In the case of PLA-OA blends, the nucleation is the more dominant factor in crystallinity development. As a result, the density of the crystalline regions is larger than the neat PLA, but the size of the individual crystal is not affected significantly by adding 1% of orotic acid as nucleating agent.

The Avrami index can intrinsically vary over a large range. Nagarajan et al. reported *n* values ranging from 1.5 to 3.6 for PLA-LAK blends at different concentrations for isotherm DSC tests. PLA mixed with nucleating agents such as LAK-301 resulted in a wider range of *n* values as compared to neat PLLA. The Avrami index is directly related to the nature of crystalline growth. Theoretically, the Avrami index ranges from 1 to 4, where lower values (1–2) are indicative of two-dimensional growth with instantaneous and sporadic nucleation while higher values (3–4) denote three-dimensional growth of spherulites with just sporadic nucleation. In this research, the *n* values were between 1.2 to 1.7 in all cases, which indicated that two-dimensional growth is favored. More specifically, for the predetermined nucleation process (constant number of nuclei per area), *n* = 1 indicates a fibrillar form of growth while *n* = 2 indicates a discoid shape growth [21]. The Avrami index between 1 and 2 matched the growth of shish-kebab structures, where the fibrillar growth forms the shish initially and the platelet-shaped kebabs form later. Additionally, the minimum *n* value of *n* = 1.2 was observed on PLA-OA blends at 100 °C. This indicated that at 100 °C, the nucleation was intense and the rodlet crystal structures filled the free volume, leaving little room for the platelet structures to grow.

### 4.3. Crystallinity of Injection Molded Parts

The crystallinity of the molded parts is presented in Table 5.

A cold crystallization peak (Figure 9) was observed on neat PLA samples and PLA-OA 80 °C samples, indicating the crystallization process is not completed during the 180-s cooling time. This is expected with a 40 °C mold temperature since it is significantly lower than the glass transition of PLA (60 °C). The molecular movement is minor on the surface layers where the polymer melt came into contact with the mold cavity. Since the temperature gradient from 215 °C at melt to 40 °C for the surface layers is very large, the rest of the polymer melt solidified at a higher rate. The material stayed at the desired crystallization temperature for very short periods, resulting in 2.6% crystallinity. Similarly, PLA-OA sample fabricated with the 80 °C temperature mold was not maintained at the temperature that offers high crystallization rate as compared to the 90 °C temperature mold. The crystallization process was not completed after 180 s cooling, but the crystallized structures were robust enough to compensate for any warpage. When the mold was opened and the sample was exposed to room temperature at 25 °C, the molecular movement stopped immediately since this temperature was below the glass transition temperature of PLA. The amorphous regions were frozen into the samples and were able to further crystallize during the DSC test, resulting in the cold crystallization peak observed in the first heating scan.

Comparing the crystallinity of injection molded samples with the materials characterized at the same temperatures, it was observed that injection molded samples presented higher crystallization than under quiescent conditions. At 80 °C, the crystallization process did not initiate after 180 s into the isotherm cycle under quiescent conditions, while for 90 °C DSC scan, the crystallinity was expected to be ~3.1% under quiescent conditions after 180 s. The significant difference between injection molded samples and samples crystallized under quiescent conditions was due to the effect of shear stress introduced during the injection molding process. The effect of controlled shear stress on the crystallization behaviors during the injection molding of commercial PLA is presented in another research individually [22].

## 5. Conclusions

The main conclusions of this research are as follows:

Orotic acid is an effective nucleating agent of PLA 2500HP over a wide range of temperatures from 80 °C to 120 °C. With the addition of only 1 wt.% of OA, the total crystallinity obtained at 80 °C was 63% as compared to 26% for neat PLA. At 100 °C, the degree of crystallinity of the neat PLA and PLA-OA blends are similar. However, PLA-OA blend reduced the crystallization time from 80 min to less than 15 min. The time for samples to reach 25% crystallinity was reduced from 13 min for neat PLA to 6.5 min for PLA-OA blend.

The 1 wt. % orotic acid is also capable of reducing the incubation time of PLA crystallization. At 100 °C, the incubation time was similar between neat PLA and PLA-OA blends, but at other temperatures, orotic acid can reduce the incubation time by at least 17%.

The 1 wt.% orotic acid increased the initial crystallization rate at all isotherm temperatures by enhancing the nucleation. The fitted crystallization rate factor k increased by two orders of magnitude. Avarmi index was between 1.2 and 1.7, indicating the formation of shish-kebab structures for all samples. At the fastest crystallization rate, the major crystal structure in PLA-OA blends was rodlet crystals. As the isotherm temperature increased/decreased, more platelet-shaped crystal structures were formed.

The 1% orotic acid was effective to initiate the crystallization behavior during the injection molding process. Neat PLA cannot be fabricated into a highly crystallized product with a cycle time less than 6 min. With the addition of 1% orotic acid, the PLA-OA blend can achieve 25.3% and 58.9% at a mold temperature of 80 °C and 90 °C, respectively. The anticipated crystallinity was minimal under quiescent conditions with 3-min isotherm. This significant increase was due to the effect of shear stresses during the injection molding process.

## Figures and Tables

**Figure 1 micromachines-13-02186-f001:**
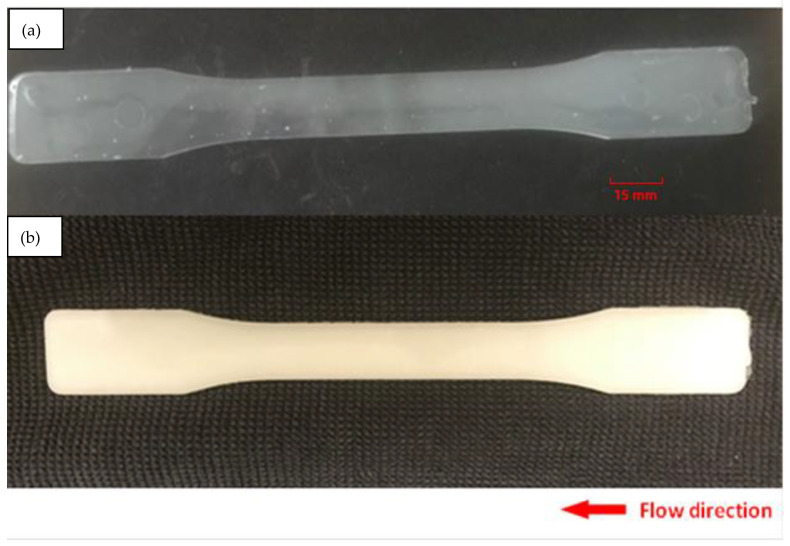
Injection molded sample: (**a**) neat PLA, (**b**) PLA-OA blend at 90 °C mold temperature.

**Figure 2 micromachines-13-02186-f002:**
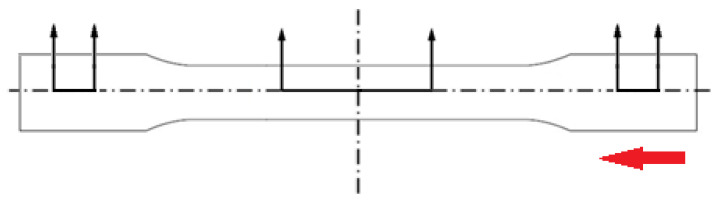
Sample preparation for DSC, a thin slice was taken from the center region and a cube sample weighing 8–10 mg was taken from that slice. The red arrow shows the flow direction.

**Figure 3 micromachines-13-02186-f003:**
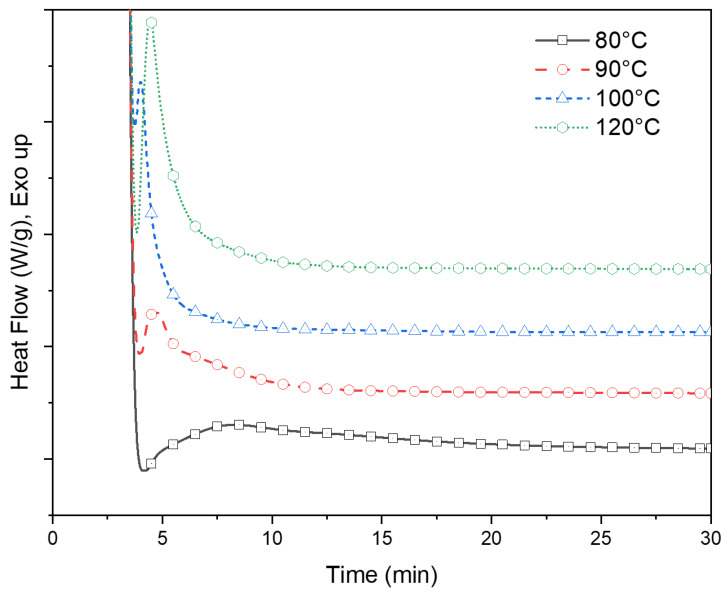
Heat flow data of PLA-OA blend isotherm tests at 80 °C, 90 °C, 100 °C, and 120 °C.

**Figure 4 micromachines-13-02186-f004:**
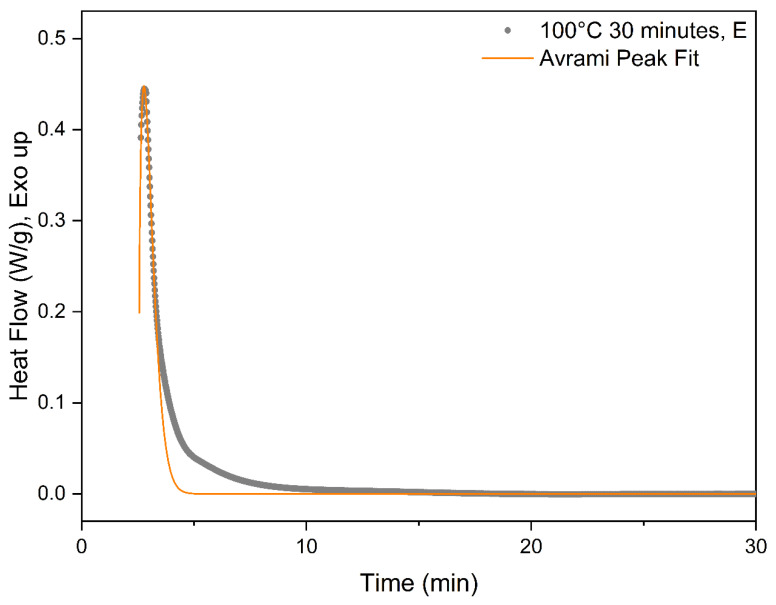
Avrami curve fitting for 100 °C DSC isotherm.

**Figure 5 micromachines-13-02186-f005:**
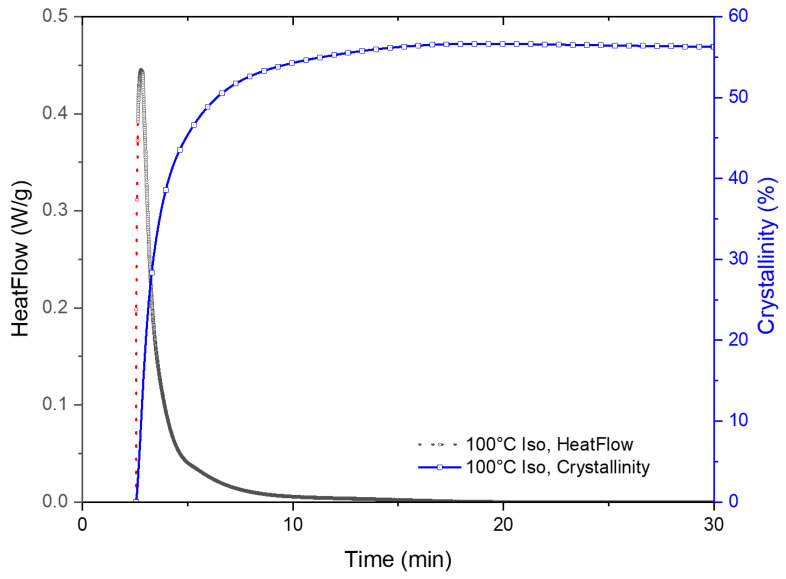
DSC curve after replacing unreliable data with fitted data. The experimental data is shown in black, and the computed data from curve fitting is presented in red. The blue curve shows the cumulative heat flow calculated by integrating the hybrid DSC isotherm curve.

**Figure 6 micromachines-13-02186-f006:**
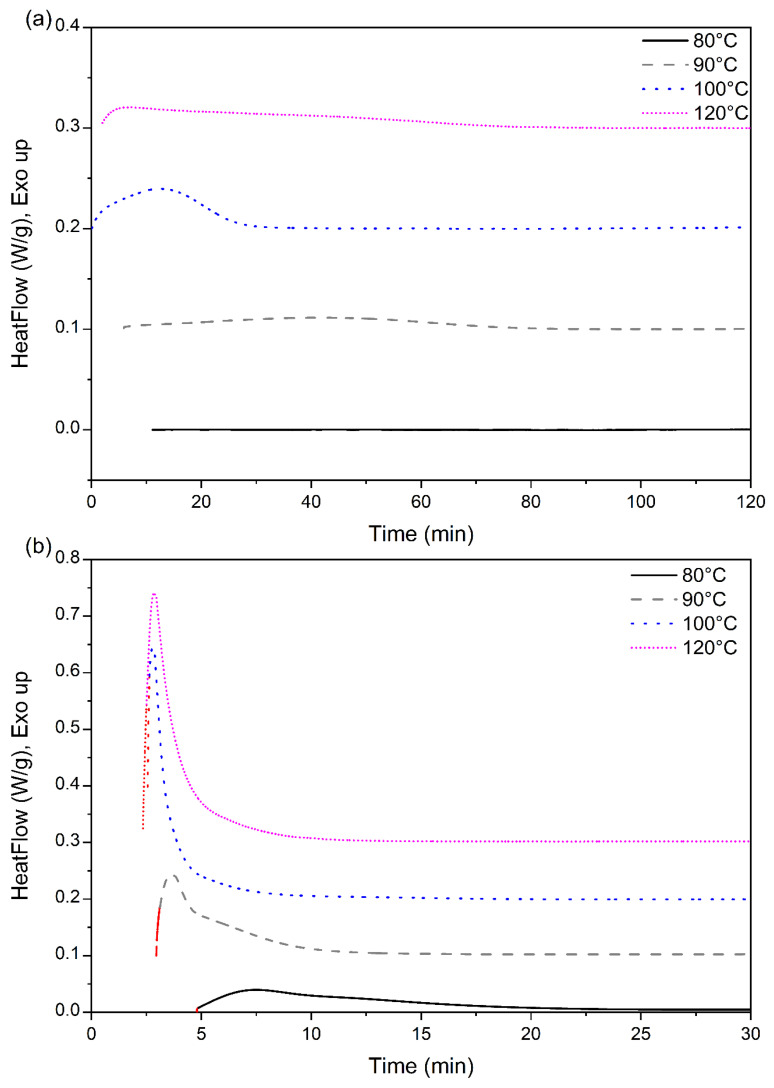
Heat flow data of (**a**) neat PLA, and (**b**) PLA-OA blend at different isotherm temperatures.

**Figure 7 micromachines-13-02186-f007:**
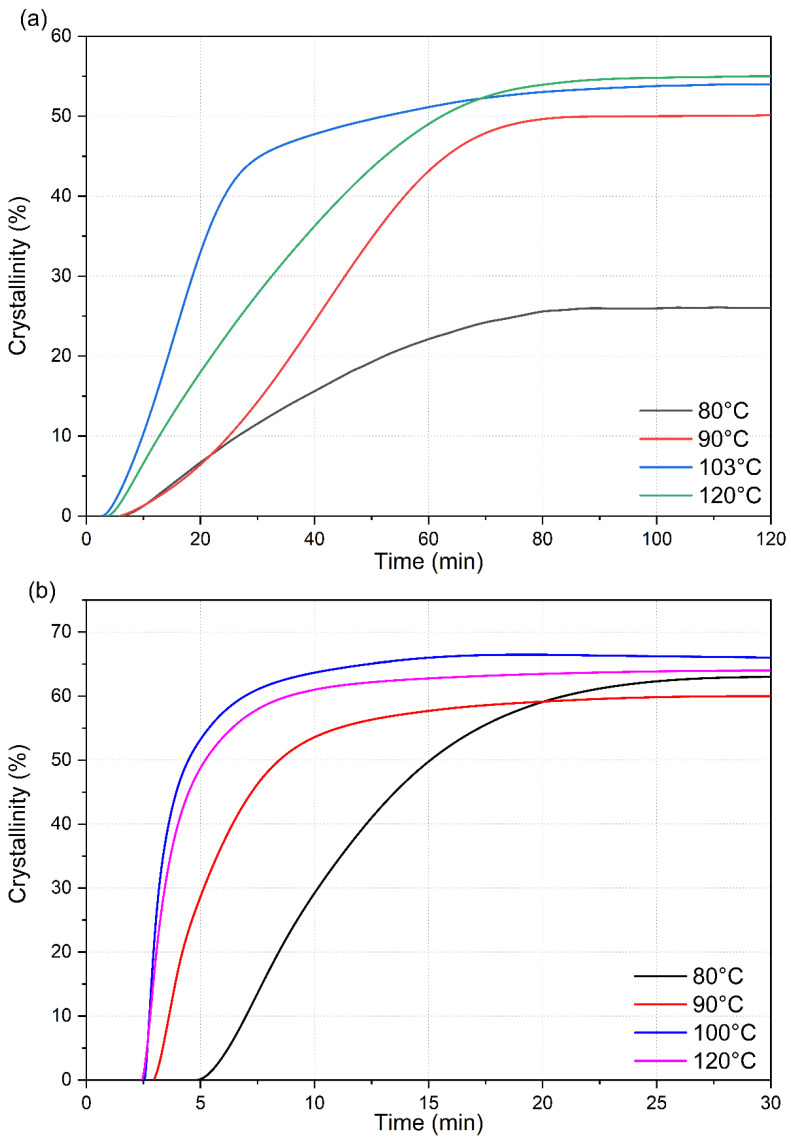
Crystallinity development of (**a**) neat PLA and (**b**) PLA-OA blend at different isotherm temperatures.

**Figure 8 micromachines-13-02186-f008:**
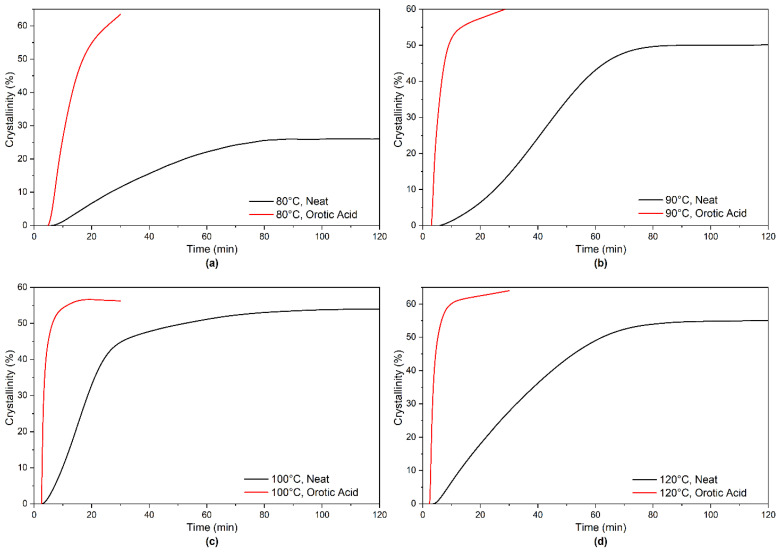
Crystallinity development curves of neat PLA and PLA-OA blend at 80–120°C.

**Figure 9 micromachines-13-02186-f009:**
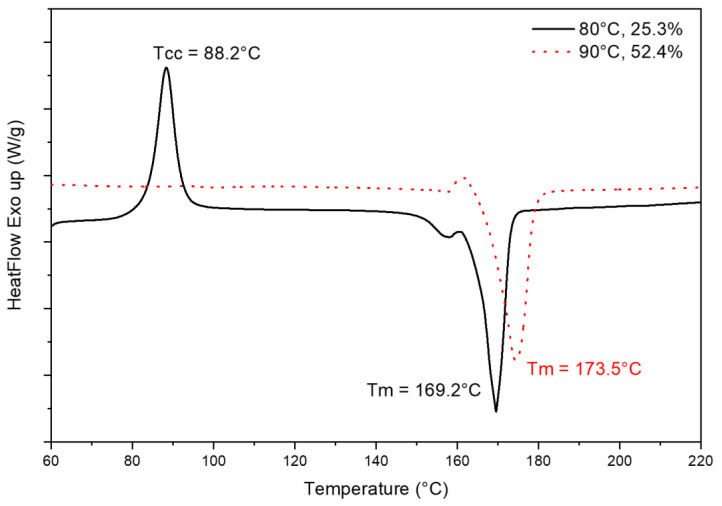
First heating cycle of DSC scans for PLA-OA blend at 80 °C and 90 °C temperature mold.

**Table 1 micromachines-13-02186-t001:** Processing Parameters for PLA Injection Molding.

Material	PLA 2500HP	PLA-OA Blend
Melt Temperature (°C)	215	
Mold Temperature (°C)	40, 60, 70, 80, 90, 95	80, 90
Injection Pressure (MPa)	64	
Packing Pressure (MPa)	36	
Cooling Time (s)	180, 360	180

**Table 2 micromachines-13-02186-t002:** Degree of crystallinity of PLA-OA blend and neat PLA as determined from the DSC melting curves.

Isotherm Temperature (°C)	Crystallinity (%) of PLA-OA Blend	Crystallinity (%) of Neat PLA
80	63	26
90	60	52
100	66	54
120	64	55

**Table 3 micromachines-13-02186-t003:** Incubation time and initial crystallization rate for neat PLA and PLA-OA blends at different isotherm temperatures.

Isotherm Temp. (°C)	Incubation Time (min)		Initial Crystallization Rate (%/min)	
	Neat PLA	PLA-OA Blend	Neat PLA	PLA-OA Blend
80	5.89	4.99	0.16	4.19
90	5.85	2.89	0.40	17.29
100	2.63	2.56	1.38	31.30
120	3.47	2.34	1.15	27.59

**Table 4 micromachines-13-02186-t004:** The *k*, *n*, $\Delta H_{c}$ values obtained from the Avrami heat flow curve fitting scheme.

Isotherm Temp. (°C)	Crystallization Rate Factor *k*		Avrami Index *n*		$\mathrm{Crystallization}\;\mathrm{Enthalpy}\;\Delta H_{c}\;(J/\mathrm{mg})$	
	Neat PLA	PLA-OA	Neat PLA	PLA-OA	Neat PLA	PLA-OA
80	1.08 × 10^−3^	-	1.6	-	0.35	-
90	1.09 × 10^−3^	0.18	1.6	1.5	0.82	0.82
100	3.73 × 10^−3^	0.57	1.7	1.2	0.90	0.82
120	3.05 × 10^−3^	0.42	1.6	1.7	0.90	0.88

**Table 5 micromachines-13-02186-t005:** Crystallinity of injection molded samples with neat PLA and PLA-OA blend.

Material and Mold Temperature	Crystallinity (%)
Neat PLA 40 °C	2.6
PLA-OA 80 °C	25.3
PLA-OA 90 °C	58.9

## Data Availability

Not applicable.

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
