# Peer review of "An Investigation on the Efficacy of Orotic Acid as a Bio-Nucleating Agent for Poly-Lactic Acid under Quiescent Condition and Injection Molding"

_micromachines, 2022, doi:10.3390/mi13122186_

Round 1
Reviewer 1 Report
This manuscript reported the effect of orotic acid (OA) as a bio-nucleating agent for PLA. However, OA is a usual agent for PLA and many works have been published about this (e.g., Prog. Polym. Sci. 2021, 37, 1657; Ind. Eng. Chem. Res. 2011, 50, 12299; Materials 2019, 12, 481). We cannot get new knowledge from this manuscript. Therefore, we do not think this paper can be published in Micromachines at current format. Some comments were addressed as below:
1. We confused the figures about injection molded samples in Pages 4 and 5. In addition, no scale bar in Fig. 1.
2. We think some descriptions lacks before the statement “demonstrated samples fabricated with neat PLA and PLA-OA blends, respectively.” (Line 126, Page 4)
3. Isothermal melting crystallization and subsequent heating process are well known to study the isothermal crystallization and melting behavior of semi-crystalline polymers. We think there is no necessary to present a typical DSC curve (Fig. 2) to show this. Moreover, Fig. 2 is not clear enough.
4. What is the unit of Hc in Table 4?
5. Fig. 9 is messy. In totally, the figures in this paper are difficult to review.
Reviewer 2 Report
The positive effect of orotic acid on the crystallization behavior of PLA was found and investigated in the present work. However, the explanation of the OA action is rare, which should be detailed in the revised manuscript.
Besides the pisctures should be modigied.
Round 2
Reviewer 1 Report
Publish as is.
Author Response
No further modification necessary.
Reviewer 2 Report
1. The effect of OA on the the crystallization behavior of PLA should be explained. Otherwise, this is just a experimental report.
2. The figures was not checked and modified. The modification should be detailed.
Author Response
COMMENT 1:
The effect of OA on the the crystallization behavior of PLA should be explained. Otherwise, this is just a experimental report.
Comment from the authors: the Authors would like to sincerely thank the Reviewer for the comment. The effect of orotic acid is thoroughly discussed in the manuscript in the aspect of increasing degree of crystallinity, reducing incubation time, increasing crystallization rate, and affecting the morphological
properties of crystalline structures in a quiescent condition crystallization process. In an injection molding process, the orotic acid is found to reduce the energy boundary to initialize the crystallization process. As a result, a lower mold temperature can be utilized to obtain high crystallinity samples. To further discuss the effect of orotic acid in the nucleation period of the process, the authors added discussion about the effect of orotic acid on the number density of crystalline regions and the sizes of the structures.
Action taken: the following section was added to line 452: “Additionally, the crystallization rate during the first 60 seconds after crystallization process initialized increased by ~2 times as compared to the average crystallization rate of the entire isotherm cycle. This indicated that the initial crystallization, with dominated by the nucleation of potential nuclei is significant. During the initial crystallization period, the number density of nuclei increases dramatically, and created potential nucleation regions to develop and grow in the later stage. After nucleation period, the more dominant behavior is the growth of crystal regions created during the initial period. In the case of PLA-OA blends, the nucleation is the more dominant factor in crystallinity development. As a result, the number density of the crystalline regions is larger than the neat PLA, but the size of the individual crystal is not affected significantly by adding 1% of orotic acid as nucleating agent.”
COMMENT 2:
The figures was not checked and modified. The modification should be detailed.
Comment from the authors: The Authors reviewed and checked the figures thoroughly and think all figures in the manuscript is accurate and readable.
Action taken: -
Round 3
